# Validating the Chinese geriatric trigger tool and analyzing adverse drug event associated risk factors in elderly Chinese patients: A retrospective review

Qiaozhi Hu⊕[☉], Zhou Qin[☉], Mei Zhan, Zhaoyan Chen, Bin Wu*, Ting Xu⊕*

Department of Pharmacy, West China Hospital, Chengdu, Sichuan, China

☉ These authors contributed equally to this work.
* binw83@qq.com (BW); tingx2009@163.com (TX)

## Abstract

### Objective

The aim was to evaluate the performance of the initial Chinese geriatric trigger tool to detect adverse drug events (ADEs) in Chinese older patients, to attempt to shorten this list for improving the efficiency of the trigger tool, and to study the incidence and characteristics of ADEs in this population.

### Methods

A sample of 25 cases was randomly selected per half a month from eligible patients who aged 60 years and older, hospitalized more than 24 hours, and discharged or died between January 1, 2015 and December 31, 2017 in West China hospital. A two-stage retrospective chart review of the included inpatients were conducted. ADEs were detected using a list of 42 triggers previously selected by an expert panel by means of a Delphi method. The number of triggers identified and ADEs detected were recorded and the positive predictive value (PPV) of each trigger was calculated to select the most efficient triggers. Several variables were recorded, including age, sex, number of diseases, length of hospital stay and so on, to analyze the risk factor of ADEs.

### Results

Among 1800 patients, 1646 positive triggers and 296 ADEs were detected in 234 (13.00%) patients. Older patients who were younger, had more medications, longer stays or more admission, and did not experience surgical operation more likely experienced ADEs. Triggers with PPV less than 5% were eliminated, which resulted in the upgraded version of Chinese geriatric trigger tool of 20 triggers with a PPV of 28.50%. This upgraded tool accounted for 99.66% of all ADEs detected.

**Data Availability Statement:** All relevant data are within the paper and its Supporting Information files.

**Funding:** The Cadre Health Care Committee of Sichuan (Project No 2018-103) supported this study. The funders had no role in study design, data collection and analysis, decision to publish, or preparation of the manuscript.

## Conclusions

The upgraded version of Chinese geriatric trigger tool was an efficient tool for identifying ADEs in Chinese older patients. Future, the trigger tool could be incorporated into routine screen systems to provide real-time identification of ADEs, thereby enabling timely clinical interventions.

## Introduction

Patient safety data from all across the globe indicate that the burden of medication-related harm is very high. The Harvard Medical Practice Study showed that medication-related injuries were the most frequent cause of adverse events and were correlated with disabling injury in about 1% of all hospitalized patients [1]. Researchers have suggested that medication-related harms account for prolonged hospital stays, cause 100,000 deaths per year and cost as much as $10 to $150 billion in the United States annually [2,3]. For these reasons, the World Health Organization (WHO) has launched its third global patient safety challenge to promote and implement actions for improving medication safety and reducing the number of preventable adverse drug events (ADEs) [4].

Older patients are more likely to experience drug-related events and have higher ADE prevalence rates compared with other age groups due to multiple co-morbid illnesses, polypharmacy, difficulty monitoring prescribed medications, and age-related changes in pharmacokinetics and pharmacodynamics [5–7]. Older adults in the United States account for 35% of all hospital stays and 53.1% of hospital ADEs [8]. Therefore, the reduction of ADEs in these vulnerable patients has become a significant safety goal in various settings [9].

Currently, various methods for identifying ADEs have been proposed, including screening voluntary reports, mining administrative databases and reviewing patient claims and medical records [10,11]. However, the vast majority of ADEs go undetected by these traditional methods, and common detection techniques have not produced consistent approaches to measure harm [12,13]. To improve medication safety and achieve the objective of reducing the number of ADEs, health care organizations should have efficient and feasible surveillance strategies to measure ADEs and monitor the results of improvement interventions.

The Global Trigger Tool (GTT), which was developed by the Institute for Healthcare Improvement in 2003, is a low-resource option for detecting adverse events at hospitals [13]. By using "triggers" to guide medical record review, GTT is more efficient in identifying adverse events compared to traditional methods [11,14]. This tool can be used in clinical practice to track and assess adverse event rates. The GTT may also be integrated into health information technology to meet the demands of rapid and real-time identification of adverse events, enable timely interventions to mitigate adverse events, and determine the effectiveness of these interventions over time [13,15]. In addition, for different types of events, groups of people or clinical settings, such as drug, perinatal, pediatric, ambulatory care settings or mental health settings, specific sets of triggers can be customized [16–18].

In China, the National Center for Adverse Drug Reactions Monitoring has established the National Adverse Drug Reaction Monitoring System, a spontaneous reporting system, to report each adverse drug reaction or adverse drug event (ADR/ADE) and to improve the data quality management of ADR/ADE reports, a normative grading criterion based on the WHO criteria. Despite the availability of this surveillance strategy, the incidence and characteristics of ADEs in Chinese patients are largely unknown, especially for specific groups of people or

medications. Therefore, based on existing triggers described in the literature and evaluated by an expert panel following a Delphi method, an initial trigger tool for older Chinese patients was developed [19]. The initial list included a total of 42 triggers divided into six categories (laboratory index, plasma concentration, antidote, clinical symptom, intervention and other) [19].

The aim of the present study was to validate the initial Chinese geriatric trigger tool in older patients in clinical practice; to improve the efficiency of the trigger tool by attempting to shorten this list in accordance with the result obtained; and to estimate the frequency and characteristics of ADEs in this population.

## Methods

### Study design and setting

This observational, retrospective study was conducted in the West China hospital of Sichuan University, a large, tertiary general teaching hospital in China. The West China hospital has 4300 beds and provides medical services to the west and south region of China [20]. There were 263, 700 inpatients, as well as 5.44 million outpatient and emergency patients in 2018; and about a quarter of all patients were geriatric patients [20]. This hospital uses electronic medical record (EMR) and bar code systems to document medication administrations. Ethics approval was obtained from the respective ethics committees at the West China Hospital of Sichuan University, China (2018–502). The institutional ethics committee waived the requirement of informed consent for this retrospective study, and all data used in this study is fully anonymized.

Eligible patients were aged ≥60 years by the official Chinese definition [21] who had a hospital stay of no less than 24hours and had been discharged or died between January 1, 2015 and December 31, 2017. Patients who were admitted to the psychiatric, rehabilitation, ambulatory surgery, and integrated traditional/ western medicine ward were excluded. A literature review revealed that the rate of geriatric inpatient ADEs is about 20–25% [22,23]. According to calculation, the sample size (N) was set as 600 cases per year [24]. After sorting by date of discharge or died, a sample of 25 cases was randomly selected from eligible patients per half a month, for a total of 1800 cases [13]. Charts for review were randomly selected from the list of eligible patients using the randomization function found at https://www.random.org/sequences/.

### Triggers

The development of the set of triggers that was used in the study has been previously reported [19]. Briefly, 51 triggers that had been identified through a detailed literature review were evaluated by an expert panel for appropriateness for geriatric patients by two-round Delphi method. The developed tool included a total of 42 triggers that were organized in the following six categories: 15 laboratory indices, 6 plasma concentrations, 13 antidotes, 6 clinical symptoms, one intervention and one other (S1 Table).

### Records review

The researchers who completed this study were trained on its methodology through participation in a similar, previous study [21]. In the current study, a two-stage review process for medical records was conducted. In the first stage, two trained clinical pharmacists (Hu and Qin) independently reviewed each medical record for the presence of any of the triggers with a limit of no more than 20 min per chart [13]. The following sections of the charts were reviewed:

medical progress notes, nursing flow sheets, medication orders, and laboratory data. Each identified trigger was recorded for further chart analysis to determine whether an associated ADE had occurred. In the second stage, one physician reviewed all the medical records with identified triggers from the first stage to determine the presence of an ADE and assign its respective category and severity. If there was a disagreement, the final decision was made based on a consensus at the study group meetings.

Besides the identified triggers and detected ADEs, the following variables were also documented: age, sex, number of diseases, length of hospital stay, number of admissions in the previous 1-year, number of medications, number of doses, and surgeries. The severity of each ADE was evaluated using the National Coordinating Council for Medication Error Reporting and Prevention Index (NCC MERP) [25]. We focused on ADEs that cause actual patient harm. Therefore, only ADEs from categories E to I were recorded: E (temporary harm to the patient requiring intervention), F (temporary harm to the patient and requiring initial or prolonged hospitalization), category G (permanent patient harm), H (intervention required to sustain life), and I (patient death). The researchers determined whether the ADEs could have been prevented using the Six-Item Screener, in which 1 indicates "definitely not preventable" and 6 indicates "definitely preventable". The ADEs with ratings greater than 4 (i.e., more than 50% likelihood preventable) were classified as preventable [26].

The positive predictive value (PPV) of each trigger was calculated using the number of identified ADEs related to this trigger divided the number of times the trigger. The PPV for the overall trigger tool was also calculated. Finally, based on a similar study and our previous study, the triggers that were found to have a rate higher than a preestablished PPV cutoff (5%) were retained in the final tool [24,27].

## Statistical analysis

The SPSS 25.0 software was used to analyze data. Descriptive statistics were calculated for patients and ADE characteristics. The following rates were calculated: ADEs per 100 admissions, ADEs per 1000 patient days, ADEs per 1000 doses, and ADEs per 1000 medications [13]. Categorical variables were summarized using frequency counts and percent, and continuous variables were presented as means with standard deviations (SD) and medians with ranges. Comparisons between groups were made using the nonparametric Mann-Whitney U test for continuous variables, and the $\chi^2$ test for categorical variables. Stepwise logistic regression was performed for variables which associated with diagnoses at a significance level of $P<0.1$ in univariate analysis [28]. Any variable significant at a level of 0.05 after regression was reported as an independent risk factor for ADEs [28]. Multicollinearity diagnosis was performed by variance inflation factors (VIF). The variables with a VIF of more than 4 were removed [29].

## Results

### Patients characteristics

A total of 1800 randomly selected patients were reviewed. Among these patients, 746 (41.44%) were female and the mean age was 69.84 years (range 60 to 101). The mean length of hospital stays varied broadly, so that the median number of lengths of stay was 8 days (range 1 to 89). The number of medications taken per patient also varied broadly with a median of 6 (range 1 to 37), and the median number of doses per patient was 39.5 (range 1 to 1731). Of the 234 (13.00%) patients with ADEs, 47 patients suffered from more than one ADE (Table 1). According to studies in Chinese hospitals, four of the most common diseases in older Chinese people are neoplasms and diseases of the circulatory, digestive, and respiratory systems [30–32],

**Table 1. Patients characteristics.**

| Characteristics | Total (n = 1800) | Patients with ADEs (n = 234) | Patients without ADEs (n = 1566) | OR | 95%CI | P |
|---|---|---|---|---|---|---|
| Sex | | | | | | |
| Male | 1054 | 154 | 900 | 1.424 | 1.068–1.900 | 0.019 |
| Female | 746 | 80 | 666 | | | |
| Age (years) | | | | | | |
| 60–74 | 1326 | 189 | 1137 | 1.585 | 1.124–2.234 | 0.009 |
| 74- | 474 | 45 | 429 | | | |
| Intensive care units | | | | | | |
| Yes | 54 | 13 | 41 | 2.188 | 1.154–4.248 | 0.017 |
| No | 1746 | 221 | 1525 | | | |
| Admission in the previous 1-year | | | | | | |
| Yes | 814 | 165 | 649 | 3.379 | 2.507–4.554 | <0.001 |
| No | 986 | 69 | 917 | | | |
| Allergic history | | | | | | |
| Yes | 193 | 24 | 169 | 0.945 | 0.601–1.484 | 0.91 |
| No | 1607 | 210 | 1397 | | | |
| Surgery performed | | | | | | |
| Yes | 660 | 32 | 628 | 0.237 | 0.161–0.348 | <0.001 |
| No | 1140 | 202 | 938 | | | |
| Type of admission | | | | | | |
| Elective | 1507 | 187 | 1320 | 0.741 | 0.524–1.050 | 0.11 |
| Emergent | 293 | 47 | 246 | | | |
| Method of admission | | | | | | |
| Wheel chair, cart or other assistance | 340 | 56 | 284 | 1.420 | 1.024–1.969 | 0.039 |
| On foot | 1460 | 178 | 1282 | | | |
| Treatment outcome | | | | | | |
| Improve or cured | 1584 | 197 | 1387 | 0.687 | 0.468–1.009 | 0.066 |
| Did not improve or died | 216 | 37 | 179 | | | |
| Antibacterial use | | | | | | |
| Yes | 577 | 80 | 497 | 1.117 | 0.836–1.494 | 0.454 |
| No | 1223 | 154 | 1069 | | | |
| Chinese patent medicine use | | | | | | |
| Yes | 223 | 37 | 186 | 1.393 | 0.950–2.044 | 0.09 |
| No | 1577 | 197 | 1380 | | | |

which are similar to our study's findings. The proportion of each diseases found in our sample was lists in S2 Table.

The univariate analyses showed there were no significant differences in allergic history, type of admission, treatment outcome, antibacterial use and Chinese patent medicine use ($P > 0.05$), whereas significant differences were identified in sex, age, length of stay, number of medical diagnoses, number of admission in the previous 1-year, number of medication and doses, method of admission, intensive care units and surgeries between patients with and without ADEs ($P \leq 0.05$) in Tables 1 and 2.

## ADEs and risk factors

Among the 234 patients with ADEs, 296 ADEs were identified, including 25 preventable ADEs. Two hundred eighty-two ADEs (95.27%) occurred during hospital stays, and 14

**Table 2. Mann-Whitney U test result of risk factors.**

| Characteristics | Total (n = 1800) | Patients with ADEs (n = 234) | Patients without ADEs (n = 1566) | Z | P |
|---|---|---|---|---|---|
| Age (years) | 69.84±8.14 | 67.95±7.55 | 70.12±8.16 | -4.125 | <0.001 |
| Length of stay | 10.19±8.94 | 15.19±12.28 | 9.44±8.07 | -7.679 | <0.001 |
| Number of medical diagnoses | 4.70±3.31 | 5.73±3.74 | 4.55±3.21 | -5.237 | <0.001 |
| Number of admissions in the previous 1-year | 1.34±3.17 | 2.33±3.07 | 1.20±3.16 | -10.001 | <0.001 |
| Drugs per patient | 7.19±5.71 | 10.00±6.26 | 6.77±5.51 | -7.938 | <0.001 |
| Doses per patient | 92.63±140.74 | 157.91±228.43 | 82.87±119.45 | -5.521 | <0.001 |

(4.73%) pre-existed as the reasons for the hospital admission. Two hundred thirty-two ADEs (78.38%) were determined to be harm category E of NCC MERP; 50 (16.89%) were category F; 13 (4.39%) were category H; and 1 (0.34%) was category I (Table 3). The calculated rate of ADEs was 16.44 per 100 admissions and 16.14 per 1000 patient days, 22.60 per 1000 medications, and 1.77 per 1000 doses.

Multicollinearity diagnostic results showed that the VIF of doses per patient was larger than 4, which indicated doses per patient had collinearity with other factors (S3 Table). Logistic regression results showed that the significant factors associated with the occurrence of ADEs were age, length of stay, surgery, number of medication and admissions ($P < 0.05$) in Table 4 (The complete result was showed in S4 Table).

## Triggers

Among the 42 triggers, 34 (80.95%) were positive and 23 (54.76%) were associated with ADEs. The result of triggers was divided into two blocks (Tables 5 and 6) to summarize the outcomes of the triggers that had PPVs more than 5% and of all other triggers with PPVs less or equal than 5% (cutoff point). A total of 1646 triggers were detected, and 343 were related to ADEs (an ADE could be identified by one or more triggers). The overall PPV of the Chinese geriatric trigger tool was 20.84%. A wide variability was found in the ADEs detected and the PPVs within the six categories. The triggers of laboratory index and antidotes allowed for more ADEs to be identified, but the plasma concentration triggers identified fewer ADEs.

The 20 triggers with PPVs accounting for more than 5% of the total were selected to become the upgraded version of Chinese geriatric trigger tool, which increased overall PPV increase to 28.50%. There was only one ADE not identified by the 20 triggers. The upgraded version of the trigger tool accounted for 99.66% of all the ADEs and 100% of the preventable ADEs (Table 5).

## Discussion

Clinicians should prioritize actions that reduce incidence of avoidable harm caused by medication in their older patients. Through integrating a literature review of existing triggers with a Delphi process, we have developed an initial list of 42 triggers for detecting ADEs in Chinese geriatric inpatients [19]. Conducting a pilot-testing of this 42-trigger tool in 1800 Chinese older patients led to the identification of 13.00% of older patients with at least one ADE, and the initial list was shortened based on the results obtained. Through use of the cutoff PPV value, less robust triggers were removed and the efficiency of the trigger tool was improved. Sixteen triggers with a PPV above 20% allowed for the detection of only 62.39% of all ADEs. The triggers with a PPV above 10% allowed for the detection of 97.38% of all ADEs but the hypokalemia trigger, which could not be substitute, would not be included. Therefore, the 20 triggers with a PPV above 5% were included into the upgraded version of the Chinese geriatric

**Table 3. Types of adverse drug events.**

| Organ /System | ADE | Total ADEs | | Preventable ADEs | |
|---|---|---|---|---|---|
| | | n | % | n | % |
| Metabolism and nutrition | Hypokalemia | 4 | 1.35% | 2 | 8.00% |
| | Hyperkalemia | 2 | 0.68% | 2 | 8.00% |
| | Hypoglycemia | 4 | 1.35% | 2 | 8.00% |
| Liver and biliary system | Hepatotoxicity/ Transaminase disorder | 25 | 8.45% | 0 | 0.00% |
| Urinary system | Nephrotoxicity/ Creatinine disorder | 3 | 1.01% | 1 | 4.00% |
| | Urinary Retention | 1 | 0.34% | 0 | 0.00% |
| Infectious | Candidiasis | 2 | 0.68% | 1 | 4.00% |
| | Infection of the upper respiratory tract | 1 | 0.34% | 0 | 0.00% |
| Musculoskeletal | Myalgia | 1 | 0.34% | 0 | 0.00% |
| Immune system | Allergy | 19 | 6.42% | 1 | 4.00% |
| Constitutional symptoms | Fever | 10 | 3.38% | 0 | 0.00% |
| | Weakness | 5 | 1.69% | 0 | 0.00% |
| | Pain | 1 | 0.34% | 0 | 0.00% |
| | Cold sweating | 1 | 0.34% | 0 | 0.00% |
| Central and peripheral nervous system | Dizziness | 4 | 1.35 | 0 | 0.00% |
| | Sleepiness | 2 | 0.68% | 0 | 0.00% |
| | Tremor | 1 | 0.34% | 0 | 0.00% |
| Gastrointestinal | Constipation | 9 | 3.04% | 0 | 0.00% |
| | Diarrhea | 22 | 7.43% | 9 | 36% |
| | Nausea | 64 | 21.62% | 0 | 0.00% |
| | Anorexia | 10 | 3.38% | 0 | 0.00% |
| | Vomiting | 22 | 7.43% | 0 | 0.00% |
| | Acute gastric mucosal Injury | 1 | 0.34% | 1 | 4.00% |
| | Abdominal distension | 1 | 0.34% | 0 | 0.00% |
| Cardiovascular | Hypotension | 6 | 2.03% | 1 | 4.00% |
| | Palpitation | 2 | 0.68% | 0 | 0.00% |
| | Bradycardia | 1 | 0.34% | 0 | 0.00% |
| | Vascular headache | 1 | 0.34% | 0 | 0.00% |
| Hematologic | Hemorrhage | 21 | 7.09% | 5 | 20% |
| | Leukopenia | 36 | 12.16% | 0 | 0.00% |
| | Thrombocytopenia | 13 | 4.39% | 0 | 0.00% |
| | Hemoglobin decline | 1 | 0.34% | 0 | 0.00% |
| Total | | 296 | 100.00% | 25 | 100% |

**Table 4. Stepwise logistic regression results of the occurrence of ADEs.**

| Variables | B | SE | Wald | P | Exp(B) | 95%CI |
|---|---|---|---|---|---|---|
| Sex (Female) | 0.293 | 0.156 | 3.518 | 0.061 | 1.340 | 0.987–1.820 |
| Age | -0.046 | 0.010 | 21.184 | 0.000 | 0.955 | 0.937–0.974 |
| Length of stay | 0.043 | 0.009 | 24.595 | 0.000 | 1.044 | 1.027–1.062 |
| Intensive care units | 0.647 | 0.378 | 2.934 | 0.087 | 1.910 | 0.911–4.006 |
| Number of admissions in the previous 1-year | 0.067 | 0.018 | 13.815 | 0.000 | 1.069 | 1.032–1.107 |
| Surgery performed | -1.252 | 0.205 | 37.301 | 0.000 | 0.286 | 0.191–0.427 |
| Method of admission (On foot) | -0.397 | 0.206 | 3.703 | 0.054 | 0.673 | 0.449–1.007 |
| Number of medications per patient | 0.047 | 0.015 | 9.787 | 0.002 | 1.048 | 1.018–1.080 |
| Constant | 0.443 | 0.683 | 0.420 | 0.517 | 1.557 | |

**Table 5. Prevalence of selected triggers and ADEs.**

| No. | Selected Triggers, n | | Total ADEs | | Preventable ADEs | |
|---|---|---|---|---|---|---|
| | | | n | PPV, % | n | PPV, % |
| Laboratory index | | | | | | |
| 1 | PTT > 100s | 2 | 2 | 100.00% | 1 | 50% |
| 2 | INR > 5 | 2 | 2 | 100.00% | 1 | 50% |
| 3 | Glucose < 2.8mmol/L | 6 | 4 | 66.67% | 2 | 33.33% |
| 4 | Rising BUN or serum creatinine greater than 2 times baseline | 10 | 2 | 20.00% | 1 | 10% |
| 5 | ALT (or AST) $\geq$3 ULN and / or ALP$\geq$2 ULN and T-BIL > 2UNL (can have abnormal INR) | 51 | 31 | 60.78% | 0 | 0.00% |
| 6 | PLT < 75×10$^9$/L | 39 | 13 | 33.33% | 0 | 0.00% |
| 7 | WBC < 3.0×10$^9$/L | 52 | 37 | 71.15% | 0 | 0.00% |
| 8 | Decrease of greater than 25% in hemoglobin or hematocrit | 9 | 4 | 44.44% | 2 | 22.22% |
| 9 | K$^+$ < 3.5mmol/L | 109 | 6 | 5.50% | 2 | 1.83% |
| 10 | K$^+$ > 5.5mmol/L | 13 | 3 | 23.08% | 3 | 100.00% |
| Antidote | | | | | | |
| 11 | Antiallergic | 70 | 16 | 22.86% | 1 | 1.43% |
| 12 | Anti-emetic | 536 | 101 | 18.84% | 0 | |
| 13 | Antidiarrheal | 31 | 16 | 51.61% | 7 | 22.58% |
| 14 | Laxative | 137 | 14 | 10.22% | 1 | 0.73% |
| 15 | Transfusion or use of blood products | 28 | 5 | 17.86% | 2 | 7.14% |
| Clinical symptom | | | | | | |
| 16 | Over sedation/hypotension | 18 | 10 | 55.56% | 1 | 5.56% |
| 17 | Rash | 19 | 14 | 73.68% | 0 | 0.00% |
| 18 | Heart rates <60/min | 3 | 2 | 66.67% | 0 | 0.00% |
| Intervention | | | | | | |
| 19 | Abrupt medication stops | 40 | 40 | 100.00% | 4 | 10.00% |
| Other | | | | | | |
| 20 | Others ADEs (ADEs not related to one of the triggers listed above) | 18 | 18 | 100.00% | 2 | 11.11% |
| Subtotal | | 1193 | 340 | 28.50% | 30 | 2.51% |

PTT, partial thromboplastin time; INR, international normalized ratio; BUN, blood urea nitrogen, ALT, alanine aminotransferase; AST, aspartate aminotransferase; T-BIL, total bilirubin; PLT, platelets; WBC, white blood cells; K$^+$, potassium; UNL, upper limit of normal; PPV, positive predictive value.

trigger tool [24,27], which covers the vast majority of ADEs, makes detection of ADEs easier and allows clinicians to identify ADEs in real time. The number of triggers in the upgraded version of Chinese geriatric trigger tool was smaller compared with other trigger tools for measuring ADEs in specific populations [6,22,27], and larger when compared to the general trigger tool for ADEs [13].

Twenty-two triggers from the initial list were eliminated based on the PPV cutoff. It is worth noting that all plasma concentration triggers were eliminated. This trigger category showed a low response rate because there was no patient conducted therapeutic drug monitoring (TDM). We attribute this to the fact that TDM has not yet become widespread in China, and plasma concentration monitoring has not served as a routine monitoring index in most Chinese older inpatients [19]. Therefore, we concluded that these triggers might had an acceptable applicability for the small numbers of patients undergoing TDM and that hospitals could add these triggers to the EMR in accordance with particular objectives or medication in the future [33]. The triggers that had a strong correlation with anesthesia did not allow for identification of any ADE or only detected one ADE. For example, flumazenil or naloxone was used to discontinue the induction and maintenance of anesthesia, but their use did not detect any

**Table 6. Prevalence of excluded triggers and ADEs.**

| No. | Not-selected Triggers, n | Total ADEs | | Preventable ADEs | |
|---|---|---|---|---|---|
| | | n | PPV, % | n | PPV, % |
| Laboratory index | | | | | |
| 1 | HGB > 170g/L(man), > 150g/L(woman) | 0 | 0 | — | 0 | — |
| 2 | $Ca^{2+}$ > 2.62 mmol/L | 4 | 0 | 0.00% | 0 | 0.00% |
| 3 | TSH < 0.27 mU/L or FT4 > 22.40 pmol/L | 1 | 0 | 0.00% | 0 | 0.00% |
| 4 | TSH > 4.2mU/L or FT4 < 12.0 pmol/L | 13 | 0 | 0.00% | 0 | 0.00% |
| 5 | Clostridium difficile positive | 0 | 0 | — | 0 | — |
| Plasma concentration | | | | | |
| 6 | Digoxin > 2 ng/ mL | 0 | 0 | — | 0 | — |
| 7 | Gentamicin or Tobramycin levels peak > 10mg/L, trough > 2mg/L | 0 | 0 | — | 0 | — |
| 8 | Cyclosporin > 300μg/mL | 0 | 0 | — | 0 | — |
| 9 | Theophylline > 20mg/L | 0 | 0 | — | 0 | — |
| 10 | Tacrolimus > 20 ng/mL | 0 | 0 | — | 0 | — |
| 11 | Voriconazole levels > 5.5mg/L | 0 | 0 | — | 0 | — |
| Antidotes | | | | | |
| 12 | Vitamin K | 29 | 1 | 3.45% | 1 | 3.45% |
| 13 | Romazicon (Flumazenil) | 117 | 0 | 0.00% | 0 | 0.00% |
| 14 | Naloxone (Narcan) | 4 | 0 | 0.00% | 0 | 0.00% |
| 15 | 50% glucose | 0 | 0 | — | 0 | — |
| 16 | Protamine | 6 | 0 | 0.00% | 0 | 0.00% |
| 17 | Epinephrine | 230 | 2 | 0.87% | 0 | 0.00% |
| 18 | Glucose injection and regular insulin | 30 | 0 | 0.00% | 0 | 0.00% |
| 19 | Insulin (regular insulin or insulin analogue) used in non-diabetics | 2 | 0 | 0.00% | 0 | 0.00% |
| Clinical symptoms | | | | | |
| 20 | Dehydration | 1 | 0 | 0.00% | 0 | 0.00% |
| 21 | Psychosis | 1 | 0 | 0.00% | 0 | 0.00% |
| 22 | Respiratory rates < 12 /min | 15 | 0 | 0.00% | 0 | 0.00% |
| Subtotal | | 426 | 3 | 0.70% | 1 | 0.23% |

HGB, hemoglobin; $Ca^{2+}$, calcium; TSH, thyroid stimulating hormone; FT4, free thyroxine.

ADEs. In addition, as China undergoes marketization and privatization, which poses numerous doctor-patient social problems such as a trust crisis [34], some emergency antidotes such as vitamin K or protamine are being used early to avoid medical disputes, although their use did not represent the occurrence of ADEs. Finally, other triggers, such as respiratory rates < 12 /min or $Ca^{2+}$ > 2.62 mmol/L also did not identify any ADEs. These findings indicate the necessity of evaluating the set of triggers for use in real clinical practice.

In our study, the incidence of geriatric ADEs was found to be similar to or lower than those found in other studies [22,23,27], which might reflect variations in local practices and study participants. Among all ADEs identified in this study, most were identified as temporary harm to the patient, findings that were similar to other studies [35,36]. We found that the older patients who experienced ADEs received more medications during their hospitalization and had longer stays, as found in previous studies [22,27,35]. This shows that the number of medications taken is an important risk factor for ADEs and underscores the need to prioritize actions to benefit this especially vulnerable population.

In addition, ADEs might more likely be found in the older patients who were younger, did not experience surgeries and had had admission within the previous 1-year. To our knowledge, there are no studies indicating that the older elderly was more likely to experience ADEs [22,27,35,37]. The younger elderly was likely to experience ADEs because they more likely to receive high-risk medications. The older patients with a greater number of admissions in our study were mostly diagnosed with malignant tumors, kidney disease, diabetes and other chronic diseases. These disorders are often treated using specialized medications including chemotherapeutic drug, anticoagulants, non-steroidal anti-inflammatory drugs, and systemic corticosteroids, all of which have been shown to be high risk for ADEs [37]. On the contrary, because surgical patients were often only administered intravenous fluid therapy during surgery, these patients were administered fewer high-risk drugs, and therefore had a lower risk of ADEs.

There are several limitations in the present study that are inherent to trigger tool methodology. First, ADE detection was based solely on a retrospective review of the medical charts. The results were dependent on the quality of the documentation, which varied across different departments and doctors. Second, though the last trigger was "other ADEs", triggers were limited in number and scope, and therefore they could not capture all ADEs. Third, we were limited in using only one hospital and its respective scope of treatment. There is still room for improvement in the upgraded version of the Chinese geriatric trigger tool, as other Chinese hospitals could customize this trigger tool according to their unique objectives and select the triggers that may be most useful at any given time for surveillance and for guiding system-level interventions such as those focused on identifying ADEs associated with a particular drug or drug group.

## Conclusion

This study which included a large sample to validate the Chinese geriatric trigger tool and investigate ADEs in Chinese older patients. Despite the limitations of this study, the upgraded version of the Chinese geriatric trigger tool has been validated to an efficient list for identifying ADEs in older patients. In our study, more than 10% of the older inpatients experienced at least one ADE, and most of these experiences caused temporary harm. The most significant factors associated with ADEs included age, the number of medications administered, the length of stay, the number of admissions and whether the patient underwent surgery. In the future, the trigger tool could be incorporated into routine screening systems to provide real-time identification of ADEs, thereby enabling initiation of timely clinical interventions.

## Supporting information

**S1 Table. Initial Chinese geriatric trigger tool.**
(DOCX)

**S2 Table. Proportion of diseases among elderly patients.**
(DOCX)

**S3 Table. Multicollinearity diagnostic result.**
(DOCX)

**S4 Table. The complete result of stepwise logistic regression.**
(DOCX)

**S1 Data.**
(RAR)

## Acknowledgments

We thank Miye Wang, who work in the information center of west China hospital, for assistance with data extraction of older patients. The authors declare that there are no conflicts of interest.

## Author Contributions

**Conceptualization:** Mei Zhan, Zhaoyan Chen.

**Data curation:** Qiaozhi Hu, Zhou Qin.

**Methodology:** Bin Wu.

**Writing – original draft:** Qiaozhi Hu.

**Writing – review & editing:** Mei Zhan, Zhaoyan Chen, Ting Xu.

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
