## [Decision Letter · Decision Letter 0]

31 Jan 2020

PONE-D-19-21999

Validating the Chinese geriatric trigger tool and analyzing adverse drug event associated risk factors in elderly Chinese patients: A retrospective review

PLOS ONE

Dear Mr Xu,

Thank you for submitting your manuscript to PLOS ONE. After careful consideration, we feel that it has merit but does not fully meet PLOS ONE’s publication criteria as it currently stands. Therefore, we invite you to submit a revised version of the manuscript that addresses the points raised during the review process.

All 3 reviewers independently concluded there were major revisions required. The reviewers have provided detailed instructions which we invite you to address. There are concerns around structure, language and, most importantly, statistics. We would appreciate receiving your revised manuscript by Mar 16 2020 11:59PM. To enhance the reproducibility of your results, we recommend that if applicable you deposit your laboratory protocols in protocols.io, where a protocol can be assigned its own identifier (DOI) such that it can be cited independently in the future. For instructions see: http://journals.plos.org/plosone/s/submission-guidelines#loc-laboratory-protocols

We look forward to receiving your revised manuscript.

Kind regards,

Katie MacLure, PhD, MSc (dist)., BSc (1st)

Academic Editor

PLOS ONE

Journal Requirements:

2. In the ethics statement in the manuscript and in the online submission form, please provide additional information about the patient records used in your retrospective study. Specifically, please ensure that you have discussed whether all data were fully anonymized before you accessed them and/or whether the IRB or ethics committee waived the requirement for informed consent. If patients provided informed written consent to have data from their medical records used in research, please include this information.

"The authors would like to thank all the researchers for their participation in the study.

The authors also would like to thank the Cadre Health Care Committee of Sichuan (2018-103), which approved the planning, execution and analysis as proposed in the grant application.

Funding: The Cadre Health Care Committee of Sichuan supported this study."

4. Please amend your manuscript to include your abstract after the title page.

Reviewers' comments:

Reviewer's Responses to Questions

**Comments to the Author**

1. Is the manuscript technically sound, and do the data support the conclusions?

Reviewer #1: Partly

Reviewer #2: Partly

Reviewer #3: Partly

2. Has the statistical analysis been performed appropriately and rigorously? 

Reviewer #1: I Don't Know

Reviewer #2: No

Reviewer #3: N/A

3. Have the authors made all data underlying the findings in their manuscript fully available?

Reviewer #1: Yes

Reviewer #2: Yes

Reviewer #3: No

4. Is the manuscript presented in an intelligible fashion and written in standard English?

Reviewer #1: No

Reviewer #2: Yes

Reviewer #3: No

5. Review Comments to the Author

Reviewer #1: Abstract:

The abstract is well-organised with the necessary information. Comments:

-Avoid using terms such as elderly by adopting term such as older adults

Methods: Describe how 1800 patients were selected

Conclusion: Why a shorter upgraded version ?

-Language use such as The aim of the study..

To attempt to shorten the list

Was calculated to use to select ..

Led to detect

METHODS

Sample size: Cite the calculation method: The sample size 67 was expanded to 600 cases per year.

Please explain what is subheading “Study Source” means

Please explain how this is possible: A sample of 25 cases was randomly selected from 74 eligible patients per half a month, for a total of 1800 cases

Unclear statement: There was 84 one “other ADEs” category, which comprised ADEs not related to one of the triggers listed above).

I understood the tool has been described in this article by the same authors: Hu Q, Qin Z, Zhan M, Wu B, Chen Z, Xu T. Development of a trigger tool for the detection of adverse drug events in Chinese geriatric inpatients using the Delphi method. Int J Clin Pharm 2019; 28. doi: 10.1007/s11096-019- 304 00871-x

-Is the sample used in this study considered as part of the presented data in the published article?

Conclusion: Try to avoid over-generalisation:

To our knowledge, this was the first study which included such a large sample to validate the Chinese 236 geriatric trigger tool and investigate ADEs in elderly Chinese elderly patients.

Conflict of interest in term of promoting the hospital unnecessarily: This hospital has been ranked second for 10 consecutive years in the composite index ranking of Chinese hospitals.

Reviewer #2: The authors have validated previously-developed prediction model for adverse drug events (ADEs). The model was upgraded by reducing the predictors (the triggers of ADEs). Higher positive predictive value was the goal of this study. I have several concerns before it can be considered for publication.

Major issues:

1. Line 55~67: Please be straightforward in ‘Study Design and Setting’. The commonly-expected information are the study design, name of hospital, country, and period of data extraction. The period information is still unavailable. No need to explain sample size estimation in this study that was already conducted, except for those subset of comparison that has no significant difference.

2. Line 123~124: Why did the authors choose significance level P<0.1, instead of P<0.05?

3. Line 81~83, 99~100; Table 1, 3~5: Please make a single table showing the trigger, unadjusted odds ratio (OR) with 95% confidence interval (CI) and P-value, adjusted odds ratio (aOR) with the same attributes, and positive predictive value (PPV) of each trigger. The OR is taken from a univariate logistic regression for each trigger while the aOR is taken from multivariate logistic regression including each trigger with variables of patient characteristics. There were a lot of significant differences in patient characteristics between patients with and without ADEs, while those were not variables of interest. These may affect association between each trigger and the outcome. Since this study aim to reduce the triggers from the original tools, only triggers with significant aOR should be included for the final model. Then both original and upgraded trigger tools should be fitted using the same configuration of multivariate logistic regression. The patient characteristics should be forced into the models either those using original or upgraded triggers. No predictor selection should be applied for the model using original triggers. This feature/predictor selection method may or may not be applied to the model using upgraded triggers. However, it is suggested to use the selection method if the authors want to reduce the triggers even more. It is also suggested to compare selection method (forward vs. backward vs. stepwise) since stepwise selection do not always give the best prediction model compared to other selection method. In the end, the PPV using all triggers can be calculated from the predicted outcomes of the models using either original or upgraded triggers. Meanwhile, the PPV should be calculated from univariate logistic regression including one trigger and patient characteristics without any selection method.

 

Minor issues:

Typos, grammatical errors, potentially-misleading statistical report, inadequate arguments in discussion, etc. are found in the manuscript. Please correct those. For example:

All lines: Please put superscripted citation after the punctuation (if available).

In line 69: Do you mean ‘Data Source’?

In line 70~71: … who had had a hospital stay of more than 24hours … (double ‘had’; miss the space in ’24hours’)

In line 74: The number of cases is a part of results. Please do not mention it in Methods.

In line 84: … of the triggers listed above)  there is no pair of the parentheses?

Line 117: Do we really need to mention the widely-used Microsoft Excel 2016? What was the critical contribution of this software for hypothesis testing?

In line 137: … and respiratory systems 29-31, results which are similar …  … and respiratory systems,29-31 which are similar …

In line 241: … underwent surgery.. (double punctuation marks)

Reviewer #3: In this manuscript, the authors aimed to evaluate the performance of the Chinese geriatric trigger tool to detect adverse drug events (ADEs) and identify the ADE-risk factors in elderly Chinese patients. The study is interesting, however, a number of issues should be addressed before its publication.

Major issues:

1. In table 1, the authors did chi-square test to evaluate the inter-group difference for categorical variables (Table 1). Can the author explain why the age grouping at 74 and 90? In addition, since the number of people over 90 is small, it might be better to merge 74-90 and 90- into one group. Otherwise, the calculated test value might be biased.

2. The authors used the stepwise logistic regression to find the risk factors for ADEs and used variance inflation factors (VIFs) to identify collinearity and remove the variables with VIF > 4. According to the author's description "stepwise logistic regression was performed for variables associated with diagnoses at a significance level of p < 0.1", there should be 13 variables (7 categorical variables and 6 continuous variables) in the initial model. It would be better to represent the process of stepwise logistic regression in the Appendix and tell the readers which variables were removed due to collinearity.

3. The results described in lines 139-143 were inconsistent with Table 3.

4. The explanation of triggers in lines 79-84, lines 159-164, and tables 4 & 5 were confusing. According to tables 4 & 5, there were 14 laboratory indices, and 7 plasma concentrations, which were different from lines 82-83. It was not clear how the authors separate 42 triggers into Table 4 and Table 5, as well as what the definition of positive was.

Minor issues:

1. There were some errors in table 1, for example,

# of patients with ADEs and # of patients without ADEs;

total number for ICU (Yes vs No) and patients without ADEs for ICU (Yes vs No.)

6. PLOS authors have the option to publish the peer review history of their article (what does this mean?). If published, this will include your full peer review and any attached files.

Reviewer #1: No

Reviewer #2: No

Reviewer #3: No

---

## [Author Response · Author response to Decision Letter 0]

9 Mar 2020

1) Reviewer #1: 

-Abstract: Conclusion: Why a shorter upgraded version?

Answer: According to the previous practice and expert opinion, some new triggers had been added to the initial list. We thought these triggers could cover most of ADEs. We thought about adding new trigger that were common in the study. The efficiency of the new triggers was inaccessible, because the denominators are unknow and the PPV of new triggers could not been obtain. In addition, too much triggers are not a good thing, the PPV of the tool may low to led to decrease efficiency. Therefore, the aim was shortening the trigger tool in this study. 

-Is the sample used in this study considered as part of the presented data in the published article?

Answer: No. The ample used in this study is bigger than published article (25 per half month in this study; 20 per half month in published article), so we conducted a re-sampling with computer.

2) Reviewer #2:

-Line 123~124: Why did the authors choose significance level P<0.1, instead of P<0.05?

Answer: Because the interplay of variables, the results of univariate analysis were different to the multivariate analysis. Therefore, in general choosing significance level P<0.1 or P<0.15 was to avoid missing some important risk factors. 

-3. Line 81~83, 99~100; Table 1, 3~5: Please make a single table showing the trigger, unadjusted odds ratio (OR) with 95% confidence interval (CI) and P-value, adjusted odds ratio (aOR) with the same attributes, and positive predictive value (PPV) of each trigger. The OR is taken from a univariate logistic regression for each trigger while the aOR is taken from multivariate logistic regression including each trigger with variables of patient characteristics. There were a lot of significant differences in patient characteristics between patients with and without ADEs, while those were not variables of interest. These may affect association between each trigger and the outcome. Since this study aim to reduce the triggers from the original tools, only triggers with significant aOR should be included for the final model. Then both original and upgraded trigger tools should be fitted using the same configuration of multivariate logistic regression. The patient characteristics should be forced into the models either those using original or upgraded triggers. No predictor selection should be applied for the model using original triggers. This feature/predictor selection method may or may not be applied to the model using upgraded triggers. However, it is suggested to use the selection method if the authors want to reduce the triggers even more. It is also suggested to compare selection method (forward vs. backward vs. stepwise) since stepwise selection do not always give the best prediction model compared to other selection method. In the end, the PPV using all triggers can be calculated from the predicted outcomes of the models using either original or upgraded triggers. Meanwhile, the PPV should be calculated from univariate logistic regression including one trigger and patient characteristics without any selection method.

Answer: The unadjusted odds ratio (OR) with 95% confidence interval (CI) and P-value have been showed in Table 1; positive predictive value (PPV) of each trigger have been showed in Tables 5 and 6. 

These triggers are derived from clinical logic to flag medical records, which alerts reviewers to initiate further in-depth investigations regarding the patient’s record to determine the presence or absence of an adverse event; for example, a trigger is a value of blood glucose lower than 28 mg/dL in a patient with oral antidiabetic or insulin, which may alert professionals to perform a more detailed record review for evidence that the patient has an associated hypoglycemia. This method can be used in practice to track and assess ADE rates. The triggers and risk factor were two different things. The characteristics of patients was the risk factor of ADEs, which could be used to the prediction of ADE. However, the triggers were just to use to detect, but not used to the prediction of ADE. The “Preventable ADEs” meant the ADEs that were preventable which judged by the research team, but not meant the trigger could predict ADE. 

PPVs = number of identified ADEs/ number of times the trigger. Therefore, the PPV was just the indicators of evaluating suitability for trigger, but had nothing to do with patients with or without ADEs. So. the OR and aOR of PPV could also not be calculated, and the aOR could not the inclusion criteria of triggers. We also could not conduct logistic regression to PPV% of each trigger. Because the results of OR and logistic regression of triggers could not get, the selection trigger was based on a similar study and our previous study, the triggers that were found to have a rate higher than a preestablished PPV cutoff (5%) were retained in the final tool.

3) Reviewer #3:

-According to the author's description "stepwise logistic regression was performed for variables associated with diagnoses at a significance level of p < 0.1", there should be 13 variables (7 categorical variables and 6 continuous variables) in the initial model. It would be better to represent the process of stepwise logistic regression in the Appendix and tell the readers which variables were removed due to collinearity.

Answer: the “Admission in the previous 1-year” and “Number of admissions in the previous 1-year” were the same thing. So, there were 12 variables (6 categorical variables and 6 continuous variables) in the initial model. The complete results of stepwise logistic regression and collinearity were showed in S3 and S4 Tables 

-It was not clear how the authors separate 42 triggers into Table 4 and Table 5, as well as what the definition of positive was.

Answer: The triggers which PPVs%＞5 were showed in Table 4 (now is Table 5), and these triggers were retained in the final tool. The triggers which PPVs＜5 were showed in Table 5 (now is Table 6)

---

## [Decision Letter · Decision Letter 1]

8 Apr 2020

Validating the Chinese geriatric trigger tool and analyzing adverse drug event associated risk factors in elderly Chinese patients: A retrospective review

PONE-D-19-21999R1

Dear Dr. Xu,

We are pleased to inform you that your manuscript has been judged scientifically suitable for publication and will be formally accepted for publication once it complies with all outstanding technical requirements.

With kind regards,

Katie MacLure, PhD, MSc (dist)., BSc (1st)

Academic Editor

PLOS ONE

Additional Editor Comments (optional):

Reviewers' comments:

Reviewer's Responses to Questions

**Comments to the Author**

1. If the authors have adequately addressed your comments raised in a previous round of review and you feel that this manuscript is now acceptable for publication, you may indicate that here to bypass the “Comments to the Author” section, enter your conflict of interest statement in the “Confidential to Editor” section, and submit your "Accept" recommendation.

Reviewer #2: All comments have been addressed

Reviewer #3: All comments have been addressed

2. Is the manuscript technically sound, and do the data support the conclusions?

Reviewer #2: Yes

Reviewer #3: Yes

3. Has the statistical analysis been performed appropriately and rigorously? 

Reviewer #2: Yes

Reviewer #3: Yes

4. Have the authors made all data underlying the findings in their manuscript fully available?

Reviewer #2: Yes

Reviewer #3: Yes

5. Is the manuscript presented in an intelligible fashion and written in standard English?

Reviewer #2: Yes

Reviewer #3: Yes

6. Review Comments to the Author

Reviewer #2: Although the authors did not answered all my major or minor questions point-by-point, I managed to find the revised parts by myself.

Reviewer #3: All comments have been addressed in the revised manuscript.

One suggestion: It is better to separate the organ/system items in Table 3, or they might get mixed up.

7. PLOS authors have the option to publish the peer review history of their article (what does this mean?). If published, this will include your full peer review and any attached files.

Reviewer #2: No

Reviewer #3: No

---

## [Editor Report · Acceptance letter]

14 Apr 2020

PONE-D-19-21999R1 

Validating the Chinese geriatric trigger tool and analyzing adverse drug event associated risk factors in elderly Chinese patients: A retrospective review 

Dear Dr. Xu:

I am pleased to inform you that your manuscript has been deemed suitable for publication in PLOS ONE. Congratulations! Your manuscript is now with our production department. 

With kind regards,

on behalf of

Dr. Katie MacLure 

Academic Editor

PLOS ONE